health and disease and epidemiology

COVID-19, SARS-CoV-2, infection rate, parameter uncertainty

**Author for correspondence:**
Steven J. Phipps
e-mail: steven@ikigairesearch.org

# Robust estimates of the true (population) infection rate for COVID-19: a backcasting approach

Steven J. Phipps[1], R. Quentin Grafton[2] and Tom Kompas[3]

[1]Ikigai Research, Hobart, Tasmania, Australia
[2]Crawford School of Public Policy, Australian National University, Canberra, Australian Capital Territory, Australia
[3]Centre of Excellence for Biosecurity Risk Analysis, University of Melbourne, Melbourne, Australia

 SJP, 0000-0001-5657-8782; RQG, 0000-0002-0048-9083; TK, 0000-0002-0665-0966

Differences in COVID-19 testing and tracing across countries, as well as changes in testing within each country over time, make it difficult to estimate the true (population) infection rate based on the confirmed number of cases obtained through RNA viral testing. We applied a backcasting approach to estimate a distribution for the true (population) cumulative number of infections (infected and recovered) for 15 developed countries. Our sample comprised countries with similar levels of medical care and with populations that have similar age distributions. Monte Carlo methods were used to robustly sample parameter uncertainty. We found a strong and statistically significant negative relationship between the proportion of the population who test positive and the implied true detection rate. Despite an overall improvement in detection rates as the pandemic has progressed, our estimates showed that, as at 31 August 2020, the true number of people to have been infected across our sample of 15 countries was 6.2 (95% CI: 4.3–10.9) times greater than the reported number of cases. In individual countries, the true number of cases exceeded the reported figure by factors that range from 2.6 (95% CI: 1.8–4.5) for South Korea to 17.5 (95% CI: 12.2–30.7) for Italy.

## 1. Introduction

COVID-19, caused by SARS-CoV-2, was declared a pandemic by the World Health Organization on 11 March 2020, after first being identified in China in December 2019. At the end of September 2020, there were more than 30 million reported cases globally and around one million reported fatalities.

Since January 2020, researchers have used the reported cases of COVID-19, detected using tests for the presence of RNA material in nasal secretions or sputum in individuals, to estimate the rate of infection within a population. In many countries, an initially inadequate number of both testing kits and testing facilities, coupled with restrictions on who could be tested, meant that the number of confirmed cases as a proportion of the total population underestimated the true (population) infection rate. Quantifying the true infection rate is an urgent health priority because the data collected are still unreliable as a means to estimate the true infection rate [1]. Multiple lines of evidence also suggest that COVID-19 may be much more widespread within the population than is suggested by the outcomes of the limited direct testing conducted to date [2–12].

A complementary approach to testing for RNA material of the virus is serological testing for antibodies for COVID-19 [13–15]. Sero-surveys provide an estimate of the number of people in the sample who have antibodies to the virus at a given point in time, providing an estimate of the level and trend of the true infection rate [15,16]. Such testing needs to be repeated regularly and for an appropriately stratified random sample of the population.

A challenge with sero-surveys is that the tests are subject to both false positives and false negatives. In the case of serological testing for COVID-19, some tests have been made available without the oversight required to ensure sufficient quality and accuracy [15,17] or have not performed adequately [15,18]. Another difficulty with serological tests is that, if the true rate of infection is relatively low (say 1% or less), then the number of false positives or false negatives may render sero-surveys unreliable as means to estimate the true infection rate [14,16]. This can be the case even if the test has a high sensitivity (proportion of those tested who have had the virus and who return a positive result, in the range of 90–97%) and a high specificity (proportion of those tested who have not been infected with the virus and who return a negative result, in the range 93–100%) [19].

A statistical approach to estimate the true number of infections is to backcast and to infer the true infection rate in the past, based on the current reported fatalities due to COVID-19. This approach has been used by Flaxman et al. [5] to estimate the true infection rates for 11 European countries, and to model the rates of infection with and without physical distancing measures. We applied our own backcasting approach to estimate the true infection rate for 15 countries, without the need to employ an epidemiological model. By comparing our estimates with the reported number of confirmed cases, we derived an implied true (population) detection rate by country. Using Monte Carlo methods to sample parameter uncertainty, 95% confidence intervals for our estimates are provided.

## 2. Methods

### 2.1. Backcasting

A backcasting method was used to estimate the true cumulative number of infections. Following Flaxman et al. [5], the time from infection to death is assumed to follow a Gamma distribution. If the mean time from infection to death is $\mu$ and the standard deviation is $s$, then the distribution of times from infection to death is assumed to follow Gamma($\alpha$, $\beta$) with $\alpha = (\mu/s)^2$ and $\beta = (s^2/\mu)$.

The Gamma distribution can be used to project the number of new daily fatalities backwards in time from the time to death to the time of initial infection. Let $N_f(t)$ be the number of new fatalities to occur on day $t$. If $f(x; \alpha, \beta)$ is the probability density function for the Gamma distribution with infection fatality rate defined by IFR, then the number of new infections estimated to have occurred on day $t'$ and to have resulted in fatalities on day $t$ is given by

$$n_i(t', t) = \frac{N_f(t) \cdot f(t - t'; \alpha, \beta)}{\text{IFR}}. \tag{2.1}$$

The estimated *total* number of new infections to have occurred on day $t'$ is, therefore, given by summing the values of $n_i(t', t)$ for all possible values of $t > t'$. This estimate is corrected because not all of the fatalities to arise from infections contracted on that day $t'$ will have occurred yet. If $t_0$ is the most recent day for which fatality statistics are available and $F(x; \alpha, \beta)$ is the cumulative distribution function for the Gamma distribution, then the estimated total number of new infections to have occurred on day $t'$ is given by

$$N_i(t') = \frac{1}{F(t_0 - t'; \alpha, \beta)} \sum_{t=t'+1}^{t_0} n_i(t', t). \tag{2.2}$$

**Table 1.** The three parameters used in the backcasting exercise: name, units, and the minimum and maximum values of the uncertainty ranges sampled.

| parameter | units | minimum | maximum |
| --- | --- | --- | --- |
| infection fatality rate (IFR) | % | 0.37 | 1.15 |
| mean incubation period | days | 4.1 | 7.0 |
| mean time from symptoms to death | days | 12.8 | 19.2 |

Equations (2.1) and (2.2) rely on the values of just three unknown parameters: (i) the mean time from infection to death; (ii) the standard deviation in the time from infection to death; and (iii) the infection fatality rate. In order to make the best use of published epidemiological data for COVID-19, we took the mean time from infection to death as being the sum of two other periods: the mean incubation period, and the mean time from development of symptoms to death. The standard deviations in each of these quantities can be estimated by following the approach of Flaxman *et al.* [5] and taking the coefficients of variation as being 0.86 and 0.45, respectively. In practice, our backcasting method is, therefore, based on the three parameters provided in table 1.

Monte Carlo methods were used to sample parameter uncertainty. An ensemble with 10 000 members was generated with random draws from within the uncertainty range for each parameter, using a Gaussian probability distribution. The evidential basis used to select the uncertainty range for each parameter is described below.

*Infection fatality rate (IFR):* The value of the infection fatality rate (IFR) remains poorly constrained, not least as calculating the IFR requires the total number of infections within the population, including asymptomatic cases, to be estimated. The IFR is also known to be highly age-dependent, varying from almost zero for younger children to potentially 25% or greater for the most elderly members of the population [20,21]. Factors such as demographics and the accuracy of reported mortality statistics can, therefore, result in regional variations in the IFR [20–22]. A recent review and meta-analysis determined that a best estimate of the population IFR is 0.68%, with a 95% confidence interval of 0.58–0.82% [22]. Of the 26 studies that were included, individual best estimates of the IFR ranged from 0.09% [23] to 1.60% [24]. When the meta-analysis was restricted to the six studies that were considered to have a low risk of bias, a best estimate of 0.76% and a 95% confidence interval of 0.37–1.15% was obtained [22]. This interval was adopted as the uncertainty range in this study.

*Mean incubation period:* Linton *et al.* [25] fitted a Gamma distribution to data for 158 confirmed cases and obtained a mean estimate for the incubation period of 6.0 days, with a 95% credible interval of 5.3–6.7 days. Examining data for 425 confirmed cases, Li *et al.* [26] fitted a lognormal distribution to a subset of cases for which detailed information was available. They obtained a mean estimate for the incubation period of 5.2 days, with a 95% confidence interval of 4.1–7.0 days. We adopted the interval 4.1–7.0 days as the uncertainty range in this study, because it encompassed the uncertainty ranges for both studies.

*Mean time from symptoms to death:* Linton *et al.* [25] fitted a Gamma distribution to data for 34 cases and obtained a mean estimate for the time from the onset of symptoms to death of 15.0 days, with a 95% credible interval of 12.8–17.5 days. Verity *et al.* [27] fitted a Gamma distribution to data for 24 deaths and 165 recoveries and obtained a mean estimate for the time from the onset of symptoms to death of 17.8 days, with a 95% credible interval of 16.9–19.2 days. We adopted the interval 12.8–19.2 days as the uncertainty range in this study, as it encompassed the uncertainty ranges for both studies.

The implied true detection rate was calculated by dividing the cumulative reported number of infections for each country by our estimates of the true cumulative number of infections.

## 2.2. Statistics

We used Spearman's rank correlation coefficient to assess correlation in this study as it is a non-parametric measure that tests for a monotonic relationship, rather than a linear relationship, between two variables. The statistical significance of Spearman's rank correlation coefficient $\rho$ was tested by calculating the *t*-statistic using

$$t = \rho\sqrt{\frac{n-2}{1-\rho^2}}.$$

(2.3)

Under the null hypothesis of statistical independence, $t$ can be assumed to be distributed as Student's $t$-distribution with $n - 2$ degrees of freedom [28]. In this study, all the tests performed were two-tailed.

## 2.3. Data

For each country, the population, the number of new infections reported each day, and the number of new fatalities reported each day were obtained from the European Centre for Disease Prevention and Control (https://www.ecdc.europa.eu/en/covid-19-pandemic). This database is updated daily, with the 14 September 2020 version used in this study.

The cumulative number of tests performed *per capita* in each country was obtained from Our World in Data (https://ourworldindata.org/coronavirus). This database is also updated daily, with the 21 September 2020 version used in this study.

A number of issues were encountered with data quality. These issues are described in appendix A.

## 2.4. Software

All the calculations presented in this study were performed using the IDL programming language, a product of Harris Geospatial Solutions (https://www.l3harrisgeospatial.com/Software-Technology/IDL). The software that we developed is available via the link provided in the data accessibility statement below.

## 3. Results

We used our backcasting approach to study the progression of the COVID-19 pandemic within 15 countries: the 11 European countries studied by Flaxman *et al.* [5], as well as Australia, Canada, South Korea and the USA. The estimated cumulative numbers of true infections within each of the 15 countries are shown in figure 1, while the implied detection rates are shown in figure 2. A summary is also presented in table 2.

The implied true detection rates have generally improved as the pandemic has progressed. Nonetheless, as at 31 August 2020, we find that the implied true detection rates are particularly low (95% CI < 10%) for Belgium, France, Italy and the UK. By contrast, the implied true detection rates are high in countries that have low incidences of COVID-19 and/or have employed widespread testing, particularly South Korea. Australia is the only country to have experienced a decline in the detection rate, with our median estimate of the detection rate stabilizing at approximately 50% during April, May and early June, before declining to 21.4% by the end of August. This decline is consistent with evidence that a resurgence of COVID-19 was accompanied by hidden transmission within communities by persons who were reluctant to get tested, even if sick, possibly because of the loss of earnings associated with self-isolation [29].

We estimated the true cumulative number of infections as at 28 March 2020 and compared our estimates with those of Flaxman *et al.* [5] for the same date (table 2). For the 11 European countries for which a comparison is possible, our median estimated infection rates tend to be slightly lower than the mean estimates of Flaxman *et al.* [5], particularly in the cases of Italy and Spain. However, our median estimate was notably higher in the case of Belgium. For all 11 countries, our 95% confidence intervals overlap with the 95% credible intervals of Flaxman *et al.* [5]. This demonstrates that our results are consistent with those generated using more complicated methods that involve the application of epidemiological models.

Our analysis covered 15 developed countries with a combined population of 817 million people. Between 28 March 2020 and 31 August 2020, we estimated that the total number of cumulative infections increased from 16.758 million (95% CI: 11.355–29.664 million) to 49.710 million (95% CI: 34.669–87.297 million). We, therefore, estimated that the fraction of the population to be infected increased from 2.05% (95% CI: 1.39–3.63%) to 6.08% (95% CI: 4.24–10.68%) over this period, with the implied true detection rate increasing from 2.6% (95% CI: 1.5–3.8%) to 16.1% (95% CI: 9.2–23.1%). These findings indicate that, on a global scale, COVID-19 is far more prevalent than is suggested by reported statistics, with the true number of infections across our sample of 15 developed countries exceeding the number of confirmed cases by a factor of 6.2 (95% CI: 4.3–10.9).

Our estimates of the implied true detection rate allowed us to explore the impact of testing on the rate of detection of COVID-19 infections in a population. We investigated the nature of this relationship at two stages during the evolution of the pandemic: an early stage (30 April 2020) and the present (31 August 2020). No significant relationship was found between the implied true detection rate and the number of tests conducted per 1000 people (figure 3). The values of Spearman's rank correlation

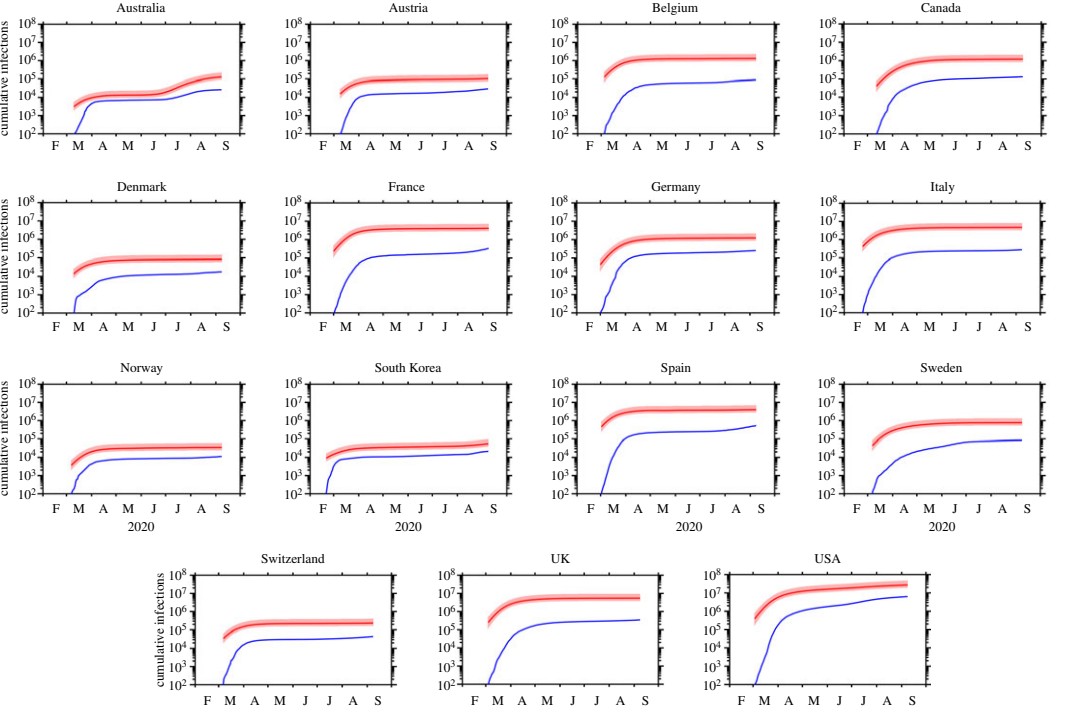

**Figure 1.** The cumulative number of COVID-19 infections for each country: the number of detected infections (blue) and the estimated true number of infections (red). For the true number, the median estimate and the 95% confidence interval are indicated by a solid line and shading respectively. The data shown for each country begins on the day that the number of detected infections first reached or exceeded 100.

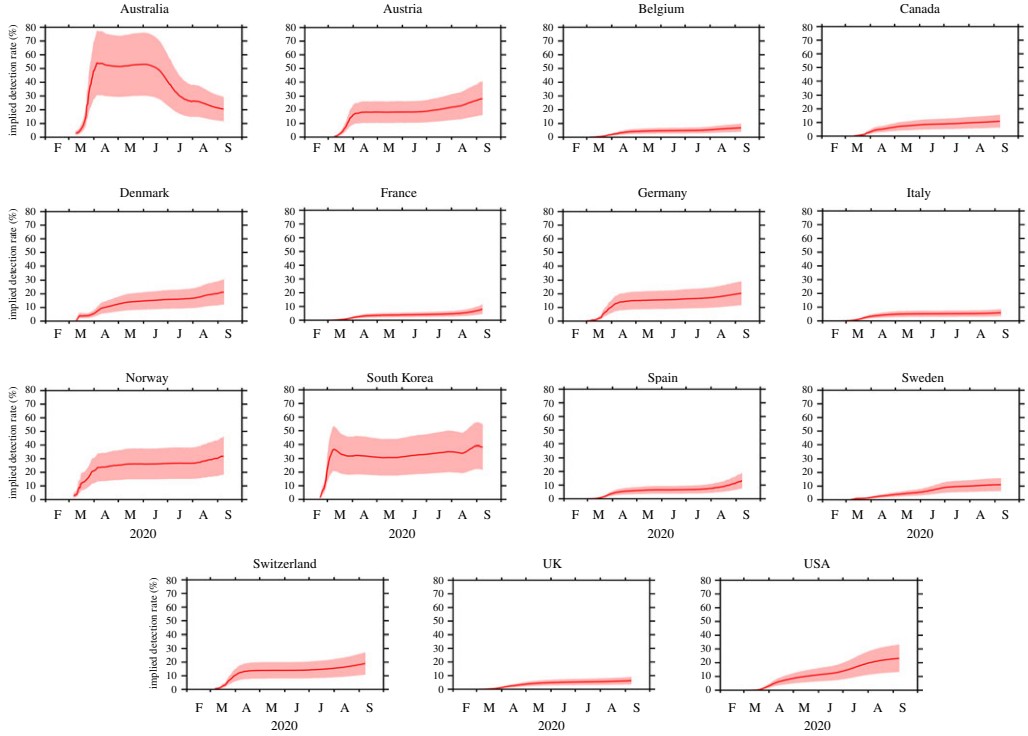

**Figure 2.** The implied detection rate for each country: the median estimate (solid line) and the 95% confidence interval (shading). The data shown for each country begins on the day that the number of detected infections first reached or exceeded 100.

**Table 2.** Statistics for each country as at 28 March 2020 and 31 August 2020: the estimated true cumulative number of infections (median and 95% confidence interval); the estimated cumulative percentage of the population to be infected (median and 95% confidence interval); the confirmed percentage of the population to have tested positive; and the implied detection rate (median and 95% confidence interval). For 28 March 2020, the estimated true cumulative number of infections according to Flaxman et al. [5] (mean and 95% credible interval) is also provided for comparison. The data shown for 'all' represents aggregated statistics for all 15 countries. Population statistics are obtained from the European Centre for Disease Prevention and Control.

| country | population (millions) | 28 March 2020 | | | | | 31 August 2020 | | | |
| --- | --- | --- | --- | --- | --- | --- | --- | --- | --- | --- |
| | | estimated cumulative infections (thousands) | % population infected this study | Flaxman et al. [5] | confirmed | implied detection rate (%) | estimated cumulative infections (thousands) | % population infected this study | confirmed | implied detection rate (%) |
| Australia | 25.203 | 9 (6–15) | 0.03 (0.02–0.06) | — | 0.01 | 38.7 (22.0–56.0) | 120 (83–211) | 0.48 (0.33–0.84) | 0.10 | 21.4 (12.2–30.8) |
| Austria | 8.859 | 55 (38–96) | 0.62 (0.42–1.09) | 1.1 (0.36–3.1) | 0.09 | 14.1 (8.0–20.5) | 101 (71–178) | 1.14 (0.80–2.01) | 0.31 | 26.9 (15.3–38.6) |
| Belgium | 11.456 | 736 (504–1301) | 6.43 (4.40–11.35) | 3.7 (1.3–9.7) | 0.11 | 1.6 (0.9–2.4) | 1307 (911–2293) | 11.41 (7.95–20.01) | 0.75 | 6.5 (3.7–9.4) |
| Canada | 37.411 | 201 (125–365) | 0.54 (0.33–0.98) | — | 0.01 | 2.3 (1.3–3.7) | 1209 (843–2121) | 3.23 (2.25–5.67) | 0.34 | 10.6 (6.0–15.2) |
| Denmark | 5.806 | 43 (30–76) | 0.74 (0.51–1.31) | 1.1 (0.40–3.1) | 0.04 | 4.7 (2.7–6.9) | 83 (58–146) | 1.43 (1.00–2.51) | 0.29 | 20.1 (11.5–28.9) |
| France | 67.013 | 2315 (1588–4087) | 3.45 (2.37–6.10) | 3.0 (1.1–7.4) | 0.05 | 1.4 (0.8–2.1) | 4087 (2849–7167) | 6.10 (4.25–10.69) | 0.41 | 6.8 (3.9–9.8) |
| Germany | 83.091 | 529 (355–935) | 0.64 (0.43–1.13) | 0.72 (0.28–1.8) | 0.06 | 9.2 (5.2–13.7) | 1232 (859–2161) | 1.48 (1.03–2.60) | 0.29 | 19.7 (11.2–28.2) |
| Italy | 60.360 | 2949 (2048–5190) | 4.89 (3.39–8.60) | 9.8 (3.2–26) | 0.14 | 2.9 (1.7–4.2) | 4688 (3269–8226) | 7.77 (5.42–13.63) | 0.44 | 5.7 (3.3–8.2) |
| Norway | 5.328 | 18 (12–32) | 0.34 (0.23–0.60) | 0.41 (0.09–1.2) | 0.07 | 19.8 (11.2–29.2) | 35 (24–61) | 0.66 (0.46–1.15) | 0.20 | 30.2 (17.2–43.3) |
| South Korea | 51.225 | 30 (21–52) | 0.06 (0.04–0.10) | — | 0.02 | 31.8 (18.1–45.7) | 51 (36–90) | 0.10 (0.07–0.17) | 0.04 | 39.1 (22.3–56.2) |
| Spain | 46.937 | 2548 (1768–4480) | 5.43 (3.77–9.54) | 15 (3.7–41) | 0.18 | 3.3 (1.9–4.7) | 3963 (2763–6956) | 8.44 (5.89–14.82) | 0.99 | 11.7 (6.7–16.8) |
| Sweden | 10.230 | 242 (163–428) | 2.36 (1.59–4.18) | 3.1 (0.85–8.4) | 0.03 | 1.4 (0.8–2.1) | 769 (536–1350) | 7.52 (5.24–13.19) | 0.82 | 11.0 (6.2–15.7) |
| Switzerland | 8.545 | 136 (93–239) | 1.59 (1.09–2.80) | 3.2 (1.3–7.6) | 0.14 | 8.9 (5.1–13.0) | 231 (161–406) | 2.71 (1.89–4.75) | 0.49 | 18.1 (10.3–26.0) |
| UK | 66.647 | 2212 (1485–3912) | 3.32 (2.23–5.87) | 2.7 (1.2–5.4) | 0.03 | 0.9 (0.5–1.4) | 5479 (3820–9614) | 8.23 (5.73–14.43) | 0.50 | 6.1 (3.5–8.8) |
| USA | 329.065 | 4737 (3121–8456) | 1.44 (0.95–2.57) | — | 0.03 | 2.2 (1.2–3.4) | 26 355 (18 387–46 319) | 8.01 (5.59–14.08) | 1.82 | 22.8 (12.9–32.6) |
| all | 817.104 | 16 758 (11 355–29 664) | 2.05 (1.39–3.63) | — | 0.05 | 2.6 (1.5–3.8) | 49 710 (34 669–87 297) | 6.08 (4.24–10.68) | 0.98 | 16.1 (9.2–23.1) |

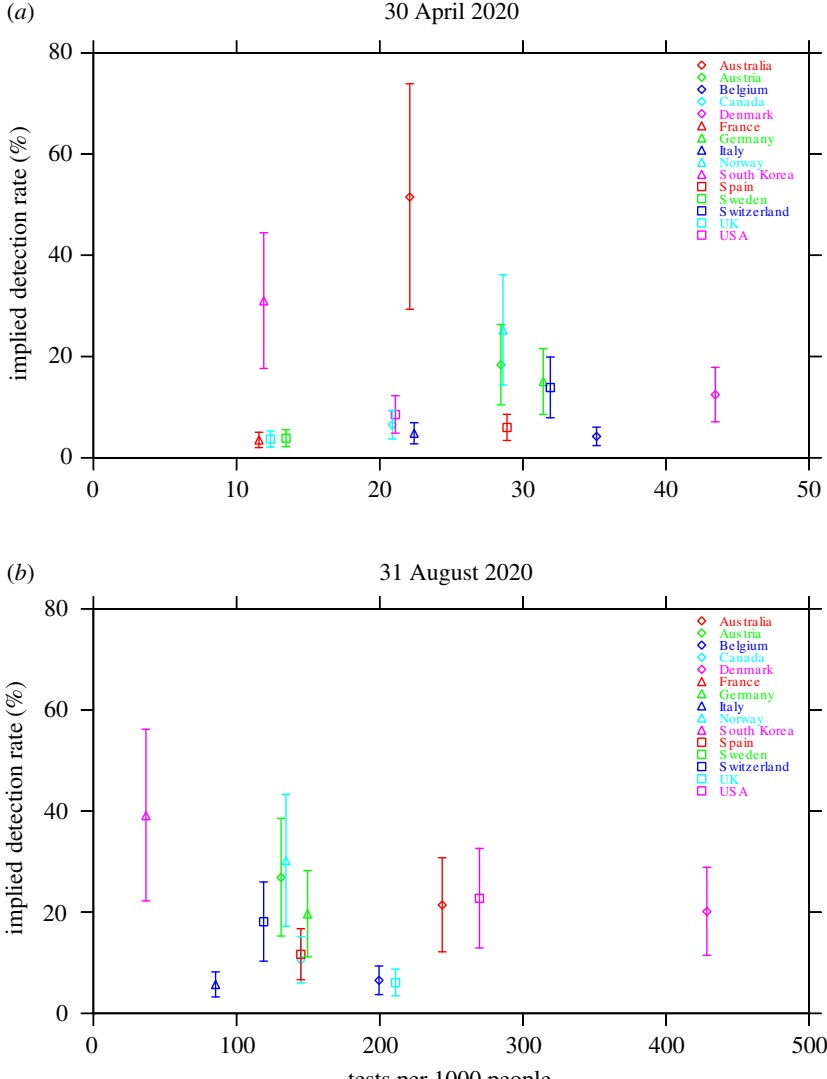

**Figure 3.** The implied true detection rate for each country versus the number of tests per 1000 people: (*a*) 30 April 2020, and (*b*) 31 August 2020. Symbols indicate the median estimates and vertical bars indicate the 95% confidence intervals. Owing to a lack of available data, values for France and Sweden are not shown in (*b*). Note that the horizontal scale is different for each panel.

coefficient are $\rho = +0.26$ ($p = 0.35$, $n = 15$) and $\rho = -0.09$ ($p = 0.78$, $n = 13$) for 30 April 2020 and 31 August 2020, respectively.

We obtained a strong negative relationship between the implied true detection rate and the fraction of tests to return a positive result, particularly during the early stage of the pandemic (figure 4*a*). The null hypothesis of no relationship is rejected using the data for 30 April 2020, with the value of Spearman's rank correlation coefficient being $\rho = -0.91$ ($p = 3.1 \times 10^{-6}$, $n = 15$). This relationship has subsequently weakened as a result of the large increase in the testing rate during the pandemic, although a negative relationship was still apparent (figure 4*b*). Using the data for 31 August 2020, the value of Spearman's rank correlation coefficient is $\rho = -0.53$ ($p = 0.061$, $n = 13$). This negative relationship means that the fraction of tests that are positive is a useful contemporaneous measure of the relative effectiveness of testing in obtaining a measure of the true (population) detection rate, particularly during the early stage of a pandemic.

## 4. Discussion

Our backcasting approach is a novel, easy-to-use and easily understood method for estimating the true (population) infection rate wherever there is reliable data on the number of fatalities attributable to COVID-19. One of its principal advantages is that it places evidence-based confidence intervals on the

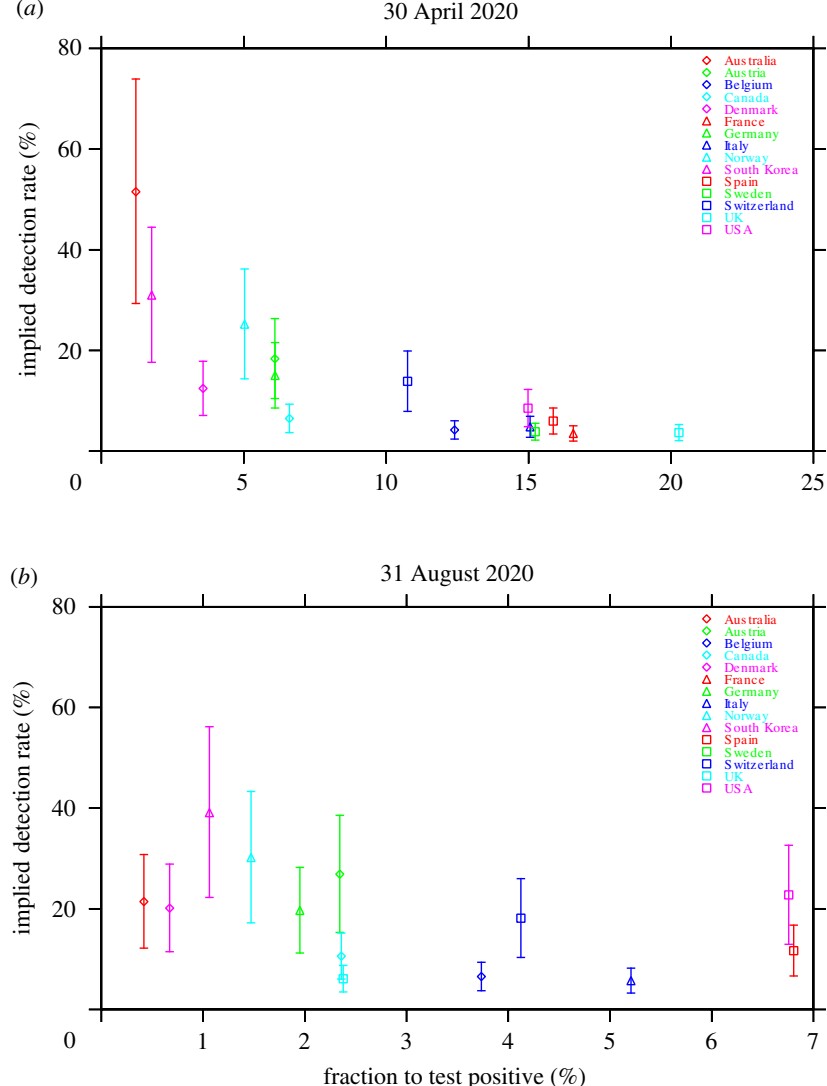

**Figure 4.** The implied true detection rate for each country versus the fraction of tests to return a positive result: (*a*) 30 April 2020, and (*b*) 31 August 2020. Symbols indicate the median estimates and vertical bars indicate the 95% confidence intervals. Owing to a lack of available data, values for France and Sweden are not shown in (*b*). Note that the horizontal scale is different for each panel.

lower and upper bounds of the number of COVID-19 infections. Unlike reported infections based on RNA tests, backcasting is not dependent on the coverage or efficacy of testing regimes, which can be very different across jurisdictions and over time.

Backcasting is scalable to a local, regional or national level, and can be readily updated on a daily basis using data that has already been reported. The relative simplicity of our approach also allows for robust sampling of parameter uncertainty. Further, our approach makes no assumptions with regard to how the number of COVID-19 infections has evolved over time. Thus, it is particularly advantageous in locations where there is little testing or limited capacity to forecast rates of infection but where there is a need, for the purposes of public health planning, for a population measure of COVID-19 infection.

Our method complements, rather than substitutes for, estimates of the true infection rate obtained through sero-surveys coupled with stratified random sampling. The difficulty with sero-surveys is that some authorized tests perform poorly. Furthermore, even if serological tests have a high sensitivity and a high specificity, sero-surveys are unreliable if used to estimate the true infection rate when the infection rate is low, because the results will be confounded by the number of false positives and false negatives.

We compared our results to Flaxman *et al.* [5] and showed that our backcasting method generates results that are consistent with those generated using more complicated methods that involve the application of epidemiological models. We also compare our estimated true cumulative infection rates with published seroprevalence studies (table 3). Our estimates are calculated on a national basis and,

**Table 3.** A comparison of published seroprevalence studies with our estimated true cumulative infection rates: the study, the country where the samples were collected, the region where the samples were collected, the dates when the samples were collected, the reported rate of seroprevalence (best estimate and 95% confidence interval), and the estimated true cumulative infection rate according to this study (median and 95% confidence interval).

| study | country | location | dates | seroprevalence (%) | this study (%) |
|---|---|---|---|---|---|
| Havers et al. [6] | USA | Western Washington State | 23 March–1 April 2020 | 1.1 (0.7–1.9) | 1.40 (0.86–2.36) |
| | | New York City metro area | 23 March–1 April 2020 | 6.9 (5.0–8.9) | 1.40 (0.86–2.36) |
| | | Louisiana | 1–8 April 2020 | 5.8 (3.9–8.2) | 2.07 (1.41–3.66) |
| | | South Florida | 6–10 April 2020 | 1.9 (1.0–3.2) | 2.35 (1.61–4.14) |
| | | Philadelphia metro area | 13–25 April 2020 | 3.2 (1.7–5.2) | 3.09 (2.15–5.44) |
| | | Missouri | 20–26 April 2020 | 2.7 (1.7–3.9) | 3.34 (2.32–5.88) |
| | | Utah | 20 April–3 May 2020 | 2.2 (1.2–3.4) | 3.52 (2.45–6.19) |
| | | San Francisco Bay area | 23–27 April 2020 | 1.0 (0.3–2.4) | 3.45 (2.40–6.07) |
| | | Connecticut | 26 April–3 May 2020 | 4.9 (3.6–6.5) | 3.68 (2.56–6.47) |
| | | Minneapolis–St Paul–St Cloud metro area | 30 April–12 May 2020 | 2.4 (1.0–4.5) | 3.97 (2.76–6.97) |
| Bendavid et al. [2] | USA | Santa Clara County, California | 3–4 April 2020 | 2.8 (1.3–4.7)[a] | 1.99 (1.35–3.52) |
| Pollán et al. [30] | Spain | nationwide | 27 April–11 May 2020 | 6.2 (5.8–6.6)[b] | 7.68 (5.35–13.48) |
| Hicks et al. [7] | Australia | nationwide | May–June 2020 | 0.28 (0.00–0.71) | 0.06 (0.04–0.10)[c] |

[a]Weighted by the authors for the population demographics of Santa Clara County.

[b]Either point-of-care test or immunoassay positive.

[c]Statistics are calculated for the period 1 May–30 June 2020.

hence, we would not expect them to replicate the results of regional assessments of seroprevalence. Nonetheless, our backcasting method generated similar results with seroprevalence studies after accounting for uncertainties. The exceptions are the New York City metro area and Louisiana in late March and early April 2020, where our estimates were significantly lower than the rates obtained from seroprevalence tests.

Backcasting has a number of advantages as a method when comparing estimated true infection rates across countries, but has three key caveats. First, the age distribution across the populations need to be broadly similar because the infection fatality rate from COVID-19 is highly dependent on age [31]. Second, the level of medical care across countries should be comparable because COVID-19 fatalities depend on access to medical services, such as the use of ventilators. Third, the infection fatality rate should be broadly constant over time, as any substantial change will introduce biases into the estimated population infection rates. Thus, appropriate cross-country comparisons impose a selection bias in terms of which countries can be included.

Backcasting complements direct testing for the virus through nasal secretions or sputum, which varies greatly between countries because of the availability of testing kits and differences in testing protocols. While countries also differ in how COVID-19 fatalities are recorded, which is a confounding factor in our method, we contend that these differences are likely to be smaller than the variations in RNA testing for the virus. Importantly, the total number of COVID-19 fatalities can be estimated if recorded fatalities are only limited to hospital fatalities by, for example, comparing the overall fatality rate to a comparable period in previous years or including a proportion of the total number of fatalities occurring outside of hospitals from COVID-19-like symptoms [32].

Complementary to backcasting are fit-for-purpose epidemiological models that are much better suited to predicting future hospitalizations and fatalities under different policy scenarios. We contend, however, that epidemiological models are not necessarily as suitable as backcasting for estimating the true infection rates because of their data requirements and modelling assumptions. Furthermore, epidemiological model parameters may not be well calibrated at a local or regional scale, and the ranges of uncertainty in the parameters may not be well understood. This may preclude robust quantification of uncertainties in model-derived estimates of infection rates.

We found, using our backcasting approach, that COVID-19 infections are far more prevalent within the populations of 15 developed countries than is indicated by the reported positive tests of RNA viral material. Our results, therefore, complement the estimated infection rates derived using a range of other techniques, including direct testing of entire communities, sero-surveys and epidemiological modelling [2–12]. A key additional advantage of our approach is that it also allows for direct comparison of infection rates over time and across countries.

An important public health finding of our study is that there is a negative relationship between the implied true detection rate and the proportion of positive viral tests for those tested for COVID-19, particularly during the early stages of the pandemic. This demonstrates both the importance and the benefit of large-scale direct testing to determine the prevalence of COVID-19 within a population. Large-scale and sufficient testing—including the testing of those who are asymptomatic—is, therefore, of critical importance to inform policy decisions about how to resource, and how to manage, the impacts of COVID-19 on public health, society and the economy.

Our backcasting approach applied to the countries in our sample implies that the true infection rate is, for the 15 countries overall, 6.2 (95% CI: 4.3–10.9) times larger than the rate implied from the number of reported cases of COVID-19. Thus, collectively for all 15 countries, we estimated that a cumulative number of 49.710 million people (95% CI: 34.669–87.297 million) had been infected with COVID-19 as at 31 August 2020, as compared to the reported total of 8.023 million. In some countries with very low implied true detection rates, such as Belgium, France, Italy and the United Kingdom, our estimates indicated that the reported number of cases, as at 31 August 2020, were likely to represent less than 10% of the true number of cases.

Most countries in the world have undertaken fewer tests per 1000 people, and have a lower capacity to test, than the 15 developed countries in our sample. Our study therefore suggests that the number of people who are infected with, or who have recovered from COVID-19, is many times greater than the reported number of cases from viral testing. A global policy implication of our finding is that rich countries should provide financial and other support to poorer countries with low levels of testing per 1000 people to support improved testing, backcasting and other methods to better measure the true (population) infection rate. In turn, improved measures of the true (population) infection rate should promote better public health decision-making in relation to COVID-19 surveillance, quarantine, contact tracing, and also the timing and stringency of government-mandated social distancing measures.

Data accessibility. The supporting data and code are deposited at Zenodo: https://zenodo.org/record/3821524.
Authors' contributions. R.Q.G. conceived of the approach. R.Q.G., S.J.P. and T.K. conceived the experiments. S.J.P. conducted the experiments and analysed the results. R.Q.G., S.J.P. and T.K. interpreted the results. All authors wrote the manuscript.
Competing interests. We declare we have no competing interests.
Funding. We received no funding for this study.
Acknowledgements. We acknowledge the European Centre for Disease Prevention and Control and Our World in Data for making data on the COVID-19 pandemic freely available. We also acknowledge use of the PyFerret program to generate the graphics in this paper. We thank Stein Ivar Steinshamn and three anonymous referees for their comments, which have substantially improved the manuscript.

# Appendix A. Data quality

A limiting factor in this study was the availability of reliable, consistent and continuous data for the number of infections, the number of fatalities and the rate of testing.

In addition to the 15 developed countries ultimately included in our sample, we also collected data on infections and fatalities for two other locations: New Zealand and Singapore. There are gaps in the European Centre for Disease Prevention and Control dataset for New Zealand, with daily data missing for 3 March 2020, 8–13 March 2020 and 16–17 March 2020. While it would have been possible for us to obtain data for these dates from alternative sources, we preferred to obtain the data for this study from a consistent source in order to minimize the risk of introducing biases into our results. We, therefore, chose to exclude New Zealand from our analysis.

In the case of Singapore, the number of reported fatalities is extremely low relative to the number of reported infections. During the three-month period from 1 June to 31 August 2020, for example, there were 22 405 reported new infections but only four reported new fatalities according to the European Centre for Disease Prevention and Control. This suggests an infection fatality rate of less than 0.02%, which lies well outside our uncertainty range of 0.37–1.15%. We hypothesize that the fatality rate is affected by the demographics of its infected population. In recent months, infections in Singapore have occurred primarily among its migrant worker population, who are young and healthy adults and who have, as a consequence, exhibited a very low mortality rate [33–35]. As the age distribution of the infected population is not representative of the population as a whole, we chose to exclude Singapore from our analysis in order to avoid introducing any unintentional biases into our results.

Obtaining reliable data for testing rates was particularly challenging, owing to a lack of consistent or continuous reporting. The difficulties of compiling statistics for testing rates are extensively documented by Our World in Data [36]. Continuous data was available for all of the 15 developed countries in our sample, for at least some stages of the pandemic. However, the continuous time series for France and Sweden end on 5 May 2020 and 5 July 2020 respectively. Values for the number of tests performed per 1000 people were generally available at daily resolution. Nevertheless, in a small number of cases, we had to use linear interpolation to estimate daily values from weekly data.

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
