## [Reviewer comments · Royal Society Open Science]

Review History

RSOS-200909.R0 (Original submission)

Review form: Reviewer 1

Is the manuscript scientifically sound in its present form?

No

Are the interpretations and conclusions justified by the results?

Yes

Is the language acceptable?

Yes

Do you have any ethical concerns with this paper?

No

Have you any concerns about statistical analyses in this paper?

No

Recommendation?

Accept with minor revision (please list in comments)

Comments to the Author(s)

The Authors have applied a backcasting method to evaluate the "true" infection rate due to Covid-19 in 15 countries, extending the work of Flaxman. The manuscript is potentially interesting for readers. However, I have some minor criticisms:

- in "Methods" Section, indicate which package was used for the statistical analysis and the implementation of Monte Carlo methods;
- in "Discussion" Section, add a comment on the comparison between the results obtained and Flaxman's results
- it must be better justified that fatalities as a result of Covid-19 are comparable among countries, a condition for applying backcasting
- it may be interesting to do some backcasting exercises, varying the parameters, for example according to age.

Review form: Reviewer 2 (Suhani Sinha)

Is the manuscript scientifically sound in its present form?

Yes

Are the interpretations and conclusions justified by the results?

Yes

Is the language acceptable?

Yes

Do you have any ethical concerns with this paper?

No

Have you any concerns about statistical analyses in this paper?

No

Recommendation?

Accept with minor revision (please list in comments)

Comments to the Author(s)

The manuscript is well written, kindly include all data and graphs in the methodology section only, not as a supplementary material at the end.

Review form: Reviewer 3 (Tao-Hsin Tung)

Is the manuscript scientifically sound in its present form?

Yes

Are the interpretations and conclusions justified by the results?

Yes

Is the language acceptable?

Yes

Do you have any ethical concerns with this paper?

No

Have you any concerns about statistical analyses in this paper?

Yes

Recommendation?

Accept with minor revision (please list in comments)

Comments to the Author(s)

The Authors propose the backcasting approach to estimates of the true (population) infection rate for COVID-19. The study designs and methods used are basically appropriate, and the interpretations of the results are reasonable. However, there are several areas where the manuscript needs to be strengthened.

1. From the epidemiologic viewpoint, there are many confounding factors in the evidenced-based researches. How the authors deal with associated confounding factors in this study?
2. The authors should point and clarify the feature and novel findings of this study.
3. Please consider the comparison with the other clinical studies using table so make clear the significance of this study.
4. How are the sensitivity and specificity in the estimates?
5. The authors should add the comments related to selection bias in this study to the perceived limitation subsection.
6. More discussion regarding the policy implications of their findings would be important for the use of methodology in health policy making.

Totally, I would like to congratulate the authors for the enthusiasm invested in this study. However, the manuscript does not reach the level of quality required for publication as original article without major revision in Royal Society Open Science.

Decision letter (RSOS-200909.R0)

Dear Dr Phipps

On behalf of the Editors, I am pleased to inform you that your Manuscript RSOS-200909 entitled "Robust estimates of the true (population) infection rate for COVID-19: A backcasting approach" has been accepted for publication in Royal Society Open Science subject to minor revision in accordance with the referee suggestions. Please find the referees' comments at the end of this email.

The reviewers and handling editors have recommended publication, but also suggest some minor revisions to your manuscript. Therefore, I invite you to respond to the comments and revise your manuscript.

- Ethics statement

- Data accessibility

It is a condition of publication that all supporting data are made available either as supplementary information or preferably in a suitable permanent repository. The data

accessibility section should state where the article's supporting data can be accessed. This section should also include details, where possible of where to access other relevant research materials such as statistical tools, protocols, software etc can be accessed. If the data has been deposited in an external repository this section should list the database, accession number and link to the DOI for all data from the article that has been made publicly available. Data sets that have been deposited in an external repository and have a DOI should also be appropriately cited in the manuscript and included in the reference list.

If you wish to submit your supporting data or code to Dryad (<http://datadryad.org/>), or modify your current submission to dryad, please use the following link:
<http://datadryad.org/submit?journalID=RSOS&manu=RSOS-200909>

- **Competing interests**

- **Authors' contributions**

- **Acknowledgements**

- **Funding statement**

Because the schedule for publication is very tight, it is a condition of publication that you submit the revised version of your manuscript before 14-Aug-2020. Please note that the revision deadline will expire at 00.00am on this date. If you do not think you will be able to meet this date please let me know immediately.

If your manuscript is newly submitted and subsequently accepted for publication, you will be asked to pay the article processing charge, unless you request a waiver and this is approved by Royal Society Publishing. You can find out more about the charges at <https://royalsocietypublishing.org/rsos/charges>. Should you have any queries, please contact openscience@royalsociety.org.

Kind regards,
Andrew Dunn
Royal Society Open Science Editorial Office

on behalf of Prof Mark Chaplain (Subject Editor)
 openscience@royalsociety.org

Associate Editor Comments to Author:

Thank you for the submission and apologies for the delay in completing review - no doubt this has been caused by the relative non-availability of many possible reviewers owing to COVID itself. In any case, the reviewers who have commented recommend the paper is largely ready for acceptance, but recommend you make a number of comparatively minor tweaks.

Reviewer comments to Author:

Reviewer: 1

Comments to the Author(s)

The Authors have applied a backcasting method to evaluate the "true" infection rate due to Covid-19 in 15 countries, extending the work of Flaxman. The manuscript is potentially interesting for readers. However, I have some minor criticisms:

- in "Methods" Section, indicate which package was used for the statistical analysis and the implementation of Monte Carlo methods;
- in "Discussion" Section, add a comment on the comparison between the results obtained and Flaxman's results
- it must be better justified that fatalities as a result of Covid-19 are comparable among countries, a condition for applying backcasting
- it may be interesting to do some backcasting exercises, varying the parameters, for example according to age.

Reviewer: 2

Comments to the Author(s)

The manuscript is well written, kindly include all data and graphs in the methodology section only, not as a supplementary material at the end.

Reviewer: 3

Comments to the Author(s)

The Authors propose the backcasting approach to estimates of the true (population) infection rate for COVID-19. The study designs and methods used are basically appropriate, and the interpretations of the results are reasonable. However, there are several areas where the manuscript needs to be strengthened.

1. From the epidemiologic viewpoint, there are many confounding factors in the evidenced-based researches. How the authors deal with associated confounding factors in this study?
2. The authors should point and clarify the feature and novel findings of this study.
3. Please consider the comparison with the other clinical studies using table so make clear the significance of this study.
4. How are the sensitivity and specificity in the estimates?
5. The authors should add the comments related to selection bias in this study to the perceived limitation subsection.
6. More discussion regarding the policy implications of their findings would be important for the use of methodology in health policy making.

Totally, I would like to congratulate the authors for the enthusiasm invested in this study. However, the manuscript does not reach the level of quality required for publication as original article without major revision in Royal Society Open Science.

Author's Response to Decision Letter for (RSOS-200909.R0)

See Appendix A.

RSOS-200909.R1 (Revision)

Review form: Reviewer 1

Is the manuscript scientifically sound in its present form?

Yes

Are the interpretations and conclusions justified by the results?

Yes

Is the language acceptable?

Yes

Do you have any ethical concerns with this paper?

No

Have you any concerns about statistical analyses in this paper?

No

Recommendation?

Accept as is

Comments to the Author(s)

I have no further comments

Review form: Reviewer 3 (Tao-Hsin Tung)

Is the manuscript scientifically sound in its present form?

Yes

Are the interpretations and conclusions justified by the results?

Yes

Is the language acceptable?

Yes

Do you have any ethical concerns with this paper?

No

Have you any concerns about statistical analyses in this paper?

No

Recommendation?

Accept as is

Comments to the Author(s)

I am pleased to accept the revised version. I have no further comments.

Decision letter (RSOS-200909.R1)

Dear Dr Phipps,

It is a pleasure to accept your manuscript entitled "Robust estimates of the true (population) infection rate for COVID-19: A backcasting approach" in its current form for publication in Royal Society Open Science. The comments of the reviewer(s) who reviewed your manuscript are included at the foot of this letter.

COVID-19 rapid publication process:

We are taking steps to expedite the publication of research relevant to the pandemic. If you wish, you can opt to have your paper published as soon as it is ready, rather than waiting for it to be published the scheduled Wednesday.

This means your paper will not be included in the weekly media round-up which the Society sends to journalists ahead of publication. However, it will still appear in the COVID-19 Publishing Collection which journalists will be directed to each week (<https://royalsocietypublishing.org/topic/special-collections/novel-coronavirus-outbreak>).

If you wish to have your paper considered for immediate publication, or to discuss further, please notify openscience_proofs@royalsociety.org and press@royalsociety.org when you respond to this email.

Kind regards,
Royal Society Open Science Editorial Office

on behalf of Prof Mark Chaplain (Subject Editor)
openscience@royalsociety.org

Reviewer comments to Author:

Reviewer: 1

Comments to the Author(s)
I have no further comments

Reviewer: 3

Comments to the Author(s)
I am pleased to accept the revised version. I have no further comments.

Appendix A

Manuscript ID RSOS-200909

Robust estimates of the true (population) infection rate for COVID-19: A backcasting approach

Steven J. Phipps, R. Quentin Grafton and Tom Kompas

We sincerely thank all three referees for their positive and constructive feedback on the manuscript in what is a difficult time for us all. We offer responses to each comment below.

In addition to revising the manuscript in response to each of the comments made by the referees, we note that we have also made the following changes:

- Updated the analysis to use the available data as at 14 September 2020.
- Updated the uncertainty ranges for our parameters to reflect current knowledge of the epidemiological characteristics of SARS-CoV-2 / COVID-19 as at September 2020.
- Other minor revisions in response to comments received on the preprint posted on medRxiv, particularly: (i) re-ordering the manuscript to follow a conventional Introduction-Methods-Results-Discussion structure, and (ii) revising Figures 3 and 4 to highlight the results for each country.

Reviewer 1

The Authors have applied a backcasting method to evaluate the "true" infection rate due to Covid-19 in 15 countries, extending the work of Flaxman. The manuscript is potentially interesting for readers. However, I have some minor criticisms:

- in "Methods" Section, indicate which package was used for the statistical analysis and the implementation of Monte Carlo methods;

We developed our own software for the backcasting method, which was written using the IDL programming language. All supporting data and code is freely available via a persistent public repository located at <https://zenodo.org/record/3821524>. We have added a new subsection to the Methods section which, in conjunction with the Data Accessibility statement, provides all of this information.

- in "Discussion" Section, add a comment on the comparison between the results obtained and Flaxman's results

We have added a sentence to the Discussion section, which discusses this comparison.

- it must be better justified that fatalities as a result of Covid-19 are comparable among countries, a condition for applying backcasting

For the current backcasting method to be comparable across countries, two factors must be consistent: (i) the age structure of the population, and (ii) the standard of, and access to, health care. Our sample of 15 developed countries was chosen on this basis. We have revised the manuscript throughout, to ensure that these points are emphasised clearly wherever relevant.

- it may be interesting to do some backcasting exercises, varying the parameters, for example according to age

The manuscript uses a Monte Carlo Method to vary all three parameters within the currently known ranges of uncertainty. Nonetheless, we agree that it would be interesting to explore the sensitivity of the parameters to the age structure of the population. Indeed, it would be necessary to this if we wished to apply the backcasting method to a greater range of countries.

We contend that, at this stage, the age dependence of the parameters is not sufficiently well understood for us to perform such an exercise in a meaningful way. This is because age dependence arises via the Infection Fatality Rate (IFR), which is currently poorly constrained even on a population-mean basis. A recent meta-analysis (Meyerowitz-Katz et al., 2020), for example, identified individual studies which derive estimates of the population-mean IFR that range from 0.09% to 1.60%. By restricting their meta-analysis to the studies that they considered to have the lowest risk of bias, they were able to derive a 95% confidence interval of 0.37-1.15%. However, this uncertainty range still represents a factor of greater than three, even in the population-mean value of the IFR.

Unfortunately, only a very limited number of studies have so far attempted to quantify the age dependence of the IFR. While it is clear that the IFR increases with age (e.g. Levin et al., 2020), the value also becomes increasingly unconstrained as age increases. Levin et al. (2020) derive a best estimate of 15% for the IFR at age 85, but their 95% prediction interval (shown in their Figure 4) spans a range from less than 5% to much more than 25% (the upper limit is not shown, as it lies outside the range of the figure). For ages above 70, Modi et al. (2020) find that the IFR can be as high as 69% (95% confidence interval: 35-100%) in the province of Bergamo, Italy. Reviewing all the data currently available, the meta-analysis of Meyerowitz-Katz et al. (2020) concludes that “There were not sufficient data in the included research to perform a meta-analysis of IFR by age.”

Thus, until better constraints on the age dependence of the parameters are available, we contend that it is appropriate to limit the backcasting exercise to countries with comparable age structures. This is exactly what we do in the manuscript, by restricting the exercise to a sample of 15 developed countries. We have revised the manuscript throughout to emphasise this caveat more clearly and thank you for highlighting this issue.

Reviewer 2

The manuscript is well written, kindly include all data and graphs in the methodology section only, not as a supplementary material at the end.

We thank the reviewer for the generous comment. We have revised the manuscript accordingly, although we note that LaTeX still chooses to place most of the figures and tables at the end of the document. We presume that this will be fixed in production, should the manuscript be accepted for final publication.

Reviewer 3

The Authors propose the backcasting approach to estimates of the true (population) infection rate for COVID-19. The study designs and methods used are basically appropriate, and the interpretations of the results are reasonable. However, there are several areas where the manuscript needs to be strengthened.

1. From the epidemiologic viewpoint, there are many confounding factors in the evidenced-based researches. How the authors deal with associated confounding factors in this study?

We acknowledge that there are a number of confounding factors: in the case of COVID-19, fatality rates can be highly dependent upon the age structure of the population, and upon the standard of

health care. As we discuss in our response to the comments of Reviewer 1 (above), we attempt to minimise the impacts of these factors on our results by restricting the analysis to a sample of 15 developed countries with comparable population age structures and health care systems. We have revised the manuscript throughout to discuss these caveats more clearly.

2.The authors should point and clarify the feature and novel findings of this study.

Our study is novel in that our backcasting method allows us to estimate the true infection rate within each country, independently of epidemiological modelling or direct serological testing. It, therefore, provides an additional, independent line of evidence for the true prevalence of COVID-19. We have revised the manuscript so that these novel features are emphasised more clearly.

3.Please consider the comparison with the other clinical studies using table so make clear the significance of this study.

We have added a new table (Table 3 in the revised manuscript), which compares the results of our backcasting method with the outcomes of serological testing. We would not expect the results to be identical, given that our backcasting method derives estimates at the national level, whereas serological testing is generally conducted on a regional basis. Nonetheless, we find that our results are consistent in almost every case, once uncertainties are taken into account.

4.How are the sensitivity and specificity in the estimates?

We have expanded the discussion of sensitivity and specificity. The revised manuscript states:

“Another difficulty with serological tests is that, if the true rate of infection is relatively low (say 1% or less), then the number of false positives or false negatives may render sero-surveys unreliable as means to estimate the true infection rate [14,16]. This can be the case even if the test has a high sensitivity (proportion of those tested who have had the virus and who return a positive result, in the range of 90–97%) and high specificity (proportion of those tested who have not been infected with the virus and who return a negative result, in the range 93–100%) [19].”

5.The authors should add the comments related to selection bias in this study to the perceived limitation subsection.

As discussed above, our study is restricted to 15 developed countries with similar population age structures and standards of health care. We have revised the manuscript throughout to emphasise this caveat more clearly.

6.More discussion regarding the policy implications of their findings would be important for the use of methodology in health policy making.

We have added a new paragraph at the end of the Discussion section, which specifically discusses the policy implications.

Totally, I would like to congratulate the authors for the enthusiasm invested in this study. However, the manuscript does not reach the level of quality required for publication as original article without major revision in Royal Society Open Science.

We thank the reviewer for the kind and constructive comments. We sincerely hope that our revisions have improved the quality of the manuscript to a satisfactory level for publication.

References

Levin AT, Meyerowitz-Katz G, Owusu-Boaitey N, Cochran KB, Walsh SP. 2020 Assessing the Age Specificity of Infection Fatality Rates for COVID-19: Systematic Review, Meta-Analysis, and Public Policy Implications. Preprint at <https://www.medrxiv.org/content/10.1101/2020.07.23.20160895v4>.

Meyerowitz-Katz G, Merone L. 2020 A systematic review and meta-analysis of published research data on COVID-19 infection-fatality rates. Preprint at <https://www.medrxiv.org/content/10.1101/2020.05.03.20089854v4>.

Modi C, Böhm V, Ferraro S, Stein G, Seljak U. 2020 How deadly is COVID-19? A rigorous analysis of excess mortality and age-dependent fatality rates in Italy. Preprint at <https://www.medrxiv.org/content/10.1101/2020.04.15.20067074v3>.